# Integrating current analyses of the breast cancer microbiome

**Sidra Sohail** [ID][1][o][¤][*], **Michael B. Burns**[1,2][o]

**1** Department of Biology Bioinformatics Program, Loyola University Chicago, Chicago, Illinois, United States of America, **2** Department of Biology, Loyola University Chicago, Chicago, Illinois, United States of America

o These authors contributed equally to this work.

¤ Current address: Department of Microbiology, Loyola University Chicago, Chicago, Illinois, United States of America

* ssohail@luc.edu

**Data Availability Statement:** Data from Hieken et al. 2016 study available at accession number PRJNA335375. Data from Urbaniak et al. 2016 study available at accession number SRP076038. Data from Chan et al. 2016 study available at accession number PRJNA314877.

## Abstract

Many cancer types have significant associations with their resident microbial communities —emerging evidence suggests that breast cancers also interact with the local tissue-associated microbiota. Microbiome research advances rapidly and analysis pipelines and databases are updated frequently. This dynamic environment makes comparative evaluations challenging. Here, we have integrated all publicly available studies related to breast cancer and the mammary microbiome in light of advances in this rapidly progressing field. Based on alpha diversity, beta diversity, proportional abundance, and statistical analyses, we observed differences between our modern analytical approaches and the original findings. We were able to classify and identify additional taxa across samples through abundance analyses and identify previously unidentified statistically significant taxa. In our updated analyses there were more taxa identified as statistically significant in comparison to the original studies' results. In the re-analysis for *The Microbiome of Aseptically Collected Human Breast Tissue in Benign and Malignant Disease* by Hieken et al., there were twelve statistically significant differentially abundant taxa identified in breast tissue microbiota in benign and invasive cancer disease states. In the re-analysis for *The Microbiota of Breast Tissue and Its Association with Breast Cancer* by Urbaniak et al., there were 18 taxa identified as statistically significant. In the re-analysis for *Characterization of the microbiome of nipple aspirate fluid of breast cancer survivors* by Chan et al., there were three genera identified as statistically significant in the skin and fluid samples. Our work has discovered that reanalyses are necessary for microbiome studies, especially older 16S studies. Through our re-analysis, we classified and identified more phyla and genera across studies, which supports the notion that reanalyses provide new insights to the microbiome field and help to assess robusticity of previously published findings by using new and updated tools and databases.

## Introduction

Breast cancer is the second leading cause of cancer death for women in the US [1]. Many cancer types have significant associations with their resident microbial communities—emerging

**Funding:** The author(s) received no specific funding for this work.

**Competing interests:** The authors have declared that no competing interests exist.

evidence suggests that breast cancers also interact with the local tissue-associated microbiota [2–5]. Studies have examined the relationship between breast cancer and its microbiome, however, the studies varied in their approaches used to evaluate these relationships. Microbiome research advances rapidly and analysis pipelines and databases are updated frequently. This dynamic environment makes inter-study comparisons and superficial evaluations challenging as no two studies are using the same standards for evaluation. The microbiome field is rapidly changing as there are different variable regions, different databases, and different pipelines available, and with this variability it is important to perform retrospective analyses to assess reproducibility of the original studies' results and to report any new findings from the updated analyses. Therefore, in this work, we are performing a retrospective analysis of existing studies on the breast cancer microbiome. Through the retrospective analysis, we report microbial taxa that were not initially identified as important by the original studies and have assessed the reproducibility of the original studies' results.

Researchers have observed the microbiota of tumor tissue, surrounding normal sites, and healthy breast tissue from non-cancer individuals [3–5], but they have not been able to translate their findings into information that can be used for breast cancer treatment or detection nor address what affect studying different variable regions has in their analysis. Within the majority of these studies, comparisons of the tumor tissue with adjacent normal tissue has revealed differences. The microbiota differ drastically with the malignant tissue showing an increased abundance of pro-inflammatory genera and a decrease in bacterial community diversity and bacterial load [3, 5]. The depleted bacterial diversity in the malignant tissue can be potentially explained by the hypoxic, inflammatory microenvironment of tumor tissue. There is a decrease in the bacterial load in advanced tumors, and also a reduction in the anti-bacterial response in the breast tumor tissue where more severe tumors have a lower abundance of innate immune receptors in breast tissue [5]. However, when evaluating the microbiota of other cancer types, the cancers harbor more diverse communities [6, 7].

In comparisons of healthy and malignant breast tissue [3], *Proteobacteria* and *Firmicutes* show increased abundance in tumor tissue [4, 5]. However, there is not a functional or clear mechanistic explanation of these differences nor any inkling of how this translates to potential treatment or improvements in diagnosis for breast cancer. Also, each study uses different bioinformatic methods, data formats, and variable regions further blurring the applicability of the results of each study to our understanding of the disease generally. Each study used a variety of different methods for their 16S rRNA gene data analysis. The beta diversity measures were calculated using an OTU table and phylogenetic tree. The observed OTU number reflects the species richness and the Shannon index reflects the species evenness. The unweighted UniFrac [8] measures differences in community presence such as whether or not there is an OTU present, and the weighted UniFrac measures differences in community abundance. The available studies [2–4] fail to address the question of what the microbial composition in the mammary microbiome entails for mammary health, and whether looking at the different variable regions in their 16S rRNA analysis has any major effects on their results. It is necessary to combine the findings from these studies with respect to the different variable regions and patient cohorts. There are a variety of taxonomic databases and each study has used a different database. As the field of microbiome research advances, the reference databases are routinely being updated and massively expanded and corrected. While the Greengenes database [9] was the best tool at the time, it is now deprecated as it has not been updated for a decade. Modern best-practices rely on up-to-date databases, including the Ribosomal Database Project (RDP) [10] and the SILVA database [11]. Through our retrospective analysis of existing studies on the breast cancer microbiome, we report new findings on microbial taxa in association with the breast cancer microbiome that were initially not discovered by the original authors.

## Results

### Hieken et al. re-analysis results

In the Hieken et al. [3] re-analysis, we performed alpha diversity, beta diversity, proportional abundance, and differential abundance analyses. In the alpha diversity analysis, we have used the observed operational taxonomic unit (OTU) number and Shannon Index to look at breast and skin tissue microbiota. The observed OTU number shows difference between the two tissues with an increased abundance of breast tissue amplicon sequence variants (ASVs) than skin tissue ASVs as shown through the observed OTU number scatter plot and box plot (Fig 1A). However, the Shannon Index does not show this difference as shown through the Shannon scatter plot and box plot (Fig 1B), and the heatmap also does not show a difference between the breast and skin ASV abundance (Fig 1C).

In the beta diversity analysis, we have used weighted and unweighted UniFrac distances which were calculated through the GUniFrac package in R [12] with the OTU table and phylogenetic tree as input. Prior to alpha and beta diversity analyses, the OTU table was rarefied to reduce confounding effects. Beta diversity analysis was performed to compare breast and skin tissue microbiota, where the unweighted UniFrac distance and weighted UniFrac distance [13] did not show a significant difference in microbial community between breast and skin tissue and the MiRKAT p-values for both unweighted and weighted UniFrac were not significant ($p > 0.05$) (Fig 1D and 1E). The microbial community between breast tissue adjacent to invasive cancer disease and breast tissue adjacent to benign disease were also compared using the unweighted and weighted UniFrac distances (Fig 1F). The unweighted UniFrac distance and

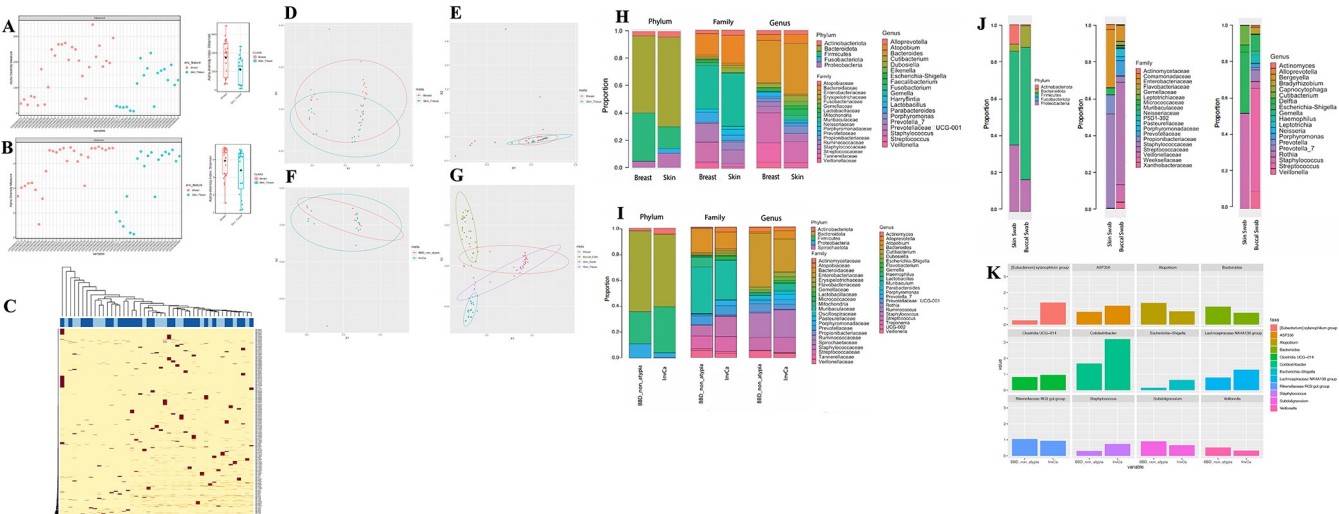

**Fig 1. Hieken study re-analysis results.** (A) The observed OTU number alpha diversity measure comparing microbiota of breast and skin tissue through a scatter plot and box plot. (B) The Shannon Index alpha diversity measure comparing microbiota of breast and skin tissue through a scatter plot and box plot. (C) Heatmap showing the ASV prevalence of breast (dark blue) and skin (light blue) tissues (columns: samples, rows: ASVs). (D) Unweighted UniFrac distance plot showing clustering of breast (red) and skin (blue) tissues. (E) Weighted UniFrac distance plot showing clustering of breast (red) and skin (blue) tissues. (F) Unweighted UniFrac distance plot showing clustering of breast tissue microbiota between benign (BBD_non_atypia) and cancer (InvCa) states. (G) PCoA plot of unweighted UniFrac distance showing clustering of all 4 tissues where the color scheme is breast tissue (red), skin tissue (purple), skin swab (blue), and buccal swab/cells (green). (H) Proportional abundance barplots generated from our DADA2 analysis of the Hieken data showing taxonomic composition of breast and skin tissue microbiota at the phylum, family, and genus levels. (I) Proportional abundance barplots generated from our DADA2 analysis of the Hieken data showing taxonomic composition of benign (BBD_non_atypia) and cancer (InvCa) samples from breast tissue at the phylum, family, and genus levels. (J) Proportional abundance barplots generated from our DADA2 analysis of the Hieken data showing taxonomic composition of buccal swab and skin swab microbiota at the phylum, family, and genus levels. (K) Differential taxa in breast tissue microbiota of benign (BBD_non_atypia) and cancer (InvCa) disease states based on the linda model.

weighted UniFrac distance did not show a significant difference in the microbial community between breast tissue adjacent to cancer disease state and breast tissue adjacent to benign disease state and the MiRKAT p-values for both unweighted and weighted UniFrac were not significant (p > 0.05). Additionally, PCoA was performed on the unweighted UniFrac distances and showed that the microbiome of the different tissue types–buccal swab, skin swab, breast tissue, and skin tissue–cluster separately from one another (Fig 1G). The buccal swab and skin swab microbiota clearly separate from each other and the other two tissues, and the skin and breast tissue microbiota are closer in space but also cluster separate from each other.

In the proportional abundance analysis, we have assessed the taxonomic composition of breast and skin tissue microbiota, breast tissue microbiota in benign and cancer disease states, and buccal and skin swab microbiota at phylum, family, and genus levels. In the Hieken et al. study, the benign disease state is referred to as BBD_non_atypia and this stands for benign breast disease without atypia, and the cancer disease state is referred to as InvCa and this stands for invasive cancer. The unweighted UniFrac distance plot (Fig 1F) and taxonomic plot (Fig 1I) use the BBD_non_atypia and InvCa abbreviations as these were extracted from the Hieken et al. study's publicly available metadata. The proportional abundance plots include the top 100 sequences in order to clearly show the taxa present at the phylum, family, and genus levels. The breast and skin tissue show similar abundances of major taxa from phyla *Actinobacteriota*, *Bacteroidota*, *Firmicutes*, and *Proteobacteria*; however, the skin tissue also shows taxa from the phylum *Fusobacteriota* (Fig 1H). The breast tissue microbiota in benign and invasive cancer disease states show similar abundances of taxa from phyla *Actinobacteriota*, *Bacteroidota*, *Firmicutes*, and *Proteobacteria*; however, the invasive cancer disease state shows a low abundance of phylum *Spirochaetota* (Fig 1I). The buccal and skin swab show similar abundances of taxa from phyla *Actinobacteriota*, *Bacteroidota*, *Firmicutes*, and *Proteobacteria* with *Fusobacteriota* found only in the skin swab (Fig 1J). At the genus level, there are clearer differences in taxonomic composition between the buccal and skin swab microbiota. The buccal swab microbiota shows a greater abundance of *Veillonella* and *Streptococcus*, whereas the skin swab microbiota shows a greater abundance of *Staphylococcus* and *Escherichia-Shigella*.

In the differential abundance analysis, taxa with prevalence of less than 10% and relative abundance of less than 0.2% were filtered out, which are the same cut-offs as the Hieken et al. study implemented for their differential abundance analysis. Using these same thresholds allows for a like-for-like comparison between the original and updated findings. In order to identify the differentially abundant taxa, we implemented a linear (lin) model for differential abundance (da) called linda which fits linear regression models on high dimensional data [14] and the linda tool is in the MicrobiomeStat package in R [15]. Based on this permutation test, there were twelve significant differentially abundant taxa identified in breast tissue microbiota in benign and invasive cancer disease states. These twelve significant differential taxa were *Eubacterium xylanophilum group*, *ASF356*, *Atopobium*, *Bacteroides*, *Clostridia UCG-014*, *Colidextribacter*, *Escherichia-Shigella*, *Lachnospiraceae NK4A136 group*, *Rikenellaceae RC9 gut group*, *Staphylococcus*, *Subdoligranulum*, and *Veillonella* where the reported p-values were unadjusted for false discovery correction. The barplots further confirm the abundances of the twelve differential taxa between the benign and malignant disease states in breast tissue (Fig 1K).

## Urbaniak et al. re-analysis results

In the Urbaniak et al. [4] re-analysis, we performed beta diversity, proportional abundance, and differential abundance analyses. In the beta diversity analysis, we performed unsupervised k-means clustering on CLR-transformed data. The clr function in the R package compositions

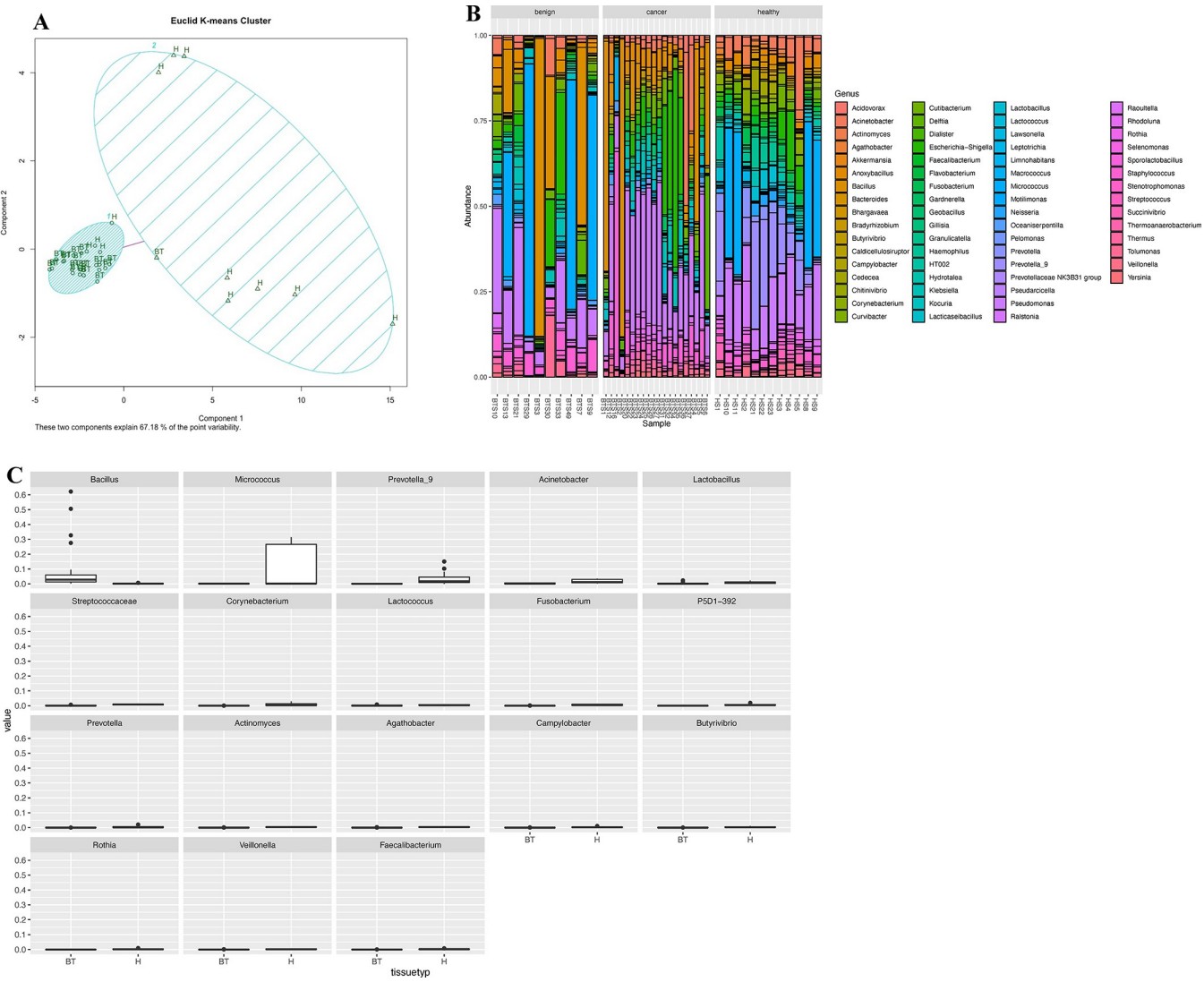

**Fig 2. Urbaniak study re-analysis results.** (A) K-means clustering plot of CLR-transformed ASV table output from the DADA2 analysis. (B) Proportional abundance plot made using the ASV table output from the DADA2 analysis and taxonomic assignment based on SILVA version 138, with benign (left), cancer (middle), and healthy (right) samples. (C) The 18 statistically significant taxa visualized through boxplots where on x-axis is healthy (right) and breast tumor (left) and y-axis is the abundance.

[16] was used to CLR-transform the data and the pam function in the R package cluster [17] was used to perform the unsupervised k-means clustering. There is a clear separation between the breast tumor (BT) cancer samples and the healthy (H) samples as shown in the clusterplot (Fig 2A) where the two components explain 67.18% of the variability.

In the proportional abundance analysis, we assessed the taxonomic composition of healthy, cancer, and benign samples at the genus level. The genus level proportional abundance plot showed a diverse population of bacteria containing 65 genera and top 100 ASVs (Fig 2B). The proportional abundance plot consists of *Pseudomonas* and *Bacillus* across the benign, cancer, and healthy samples, where *Escherichia-Shigella* was shown to be prevalent in the cancer samples.

In the differential abundance analysis, the ALDEx R package version 2 [18] was used to compare relative abundances of taxa at the genus level of CLR-transformed data. The reported

p-values from ALDEx2 are Benjamini-Hochberg corrected p-values of the Wilcoxon rank test. The ALDEx2 output was visualized through boxplots (Fig 2C) and there were 20 statistically significant ASVs and 18 statistically significant taxa of which the following genera were significantly higher in abundance in healthy samples *Acinetobacter*, *Prevotella_9*, *Lactobacillus*, *Corynebacterium*, *Lactococcus*, *Fusobacterium*, *Prevotella*, *Actinomyces*, *Agathobacter*, *Campylobacter*, *Butyrivibrio*, *Rothia*, *Veillonella*, *Faecalibacterium*, and *Micrococcus*. There was a significantly higher abundance of the following genus in breast tumor cancer samples *Bacillus*.

## Chan et al. re-analysis results

In the Chan et al. [2] re-analysis, we performed alpha diversity, beta diversity, proportional abundance, and differential abundance analyses. In the alpha diversity analysis, we have implemented the observed OTU number along with a nonparametric t-test on nipple skin (NS), nipple aspirate fluid (NAF), and post-betadine skin (PBS) microbiota. The observed OTU number shows the number of ASVs observed for the NS, NAF, and PBS tissues. The t-test assesses whether the microbial diversity is significantly different between the healthy control and cancer samples from NS, NAF, and PBS environments. The p-values were not significant for NS, NAF, and PBS samples (Fig 3A–3C).

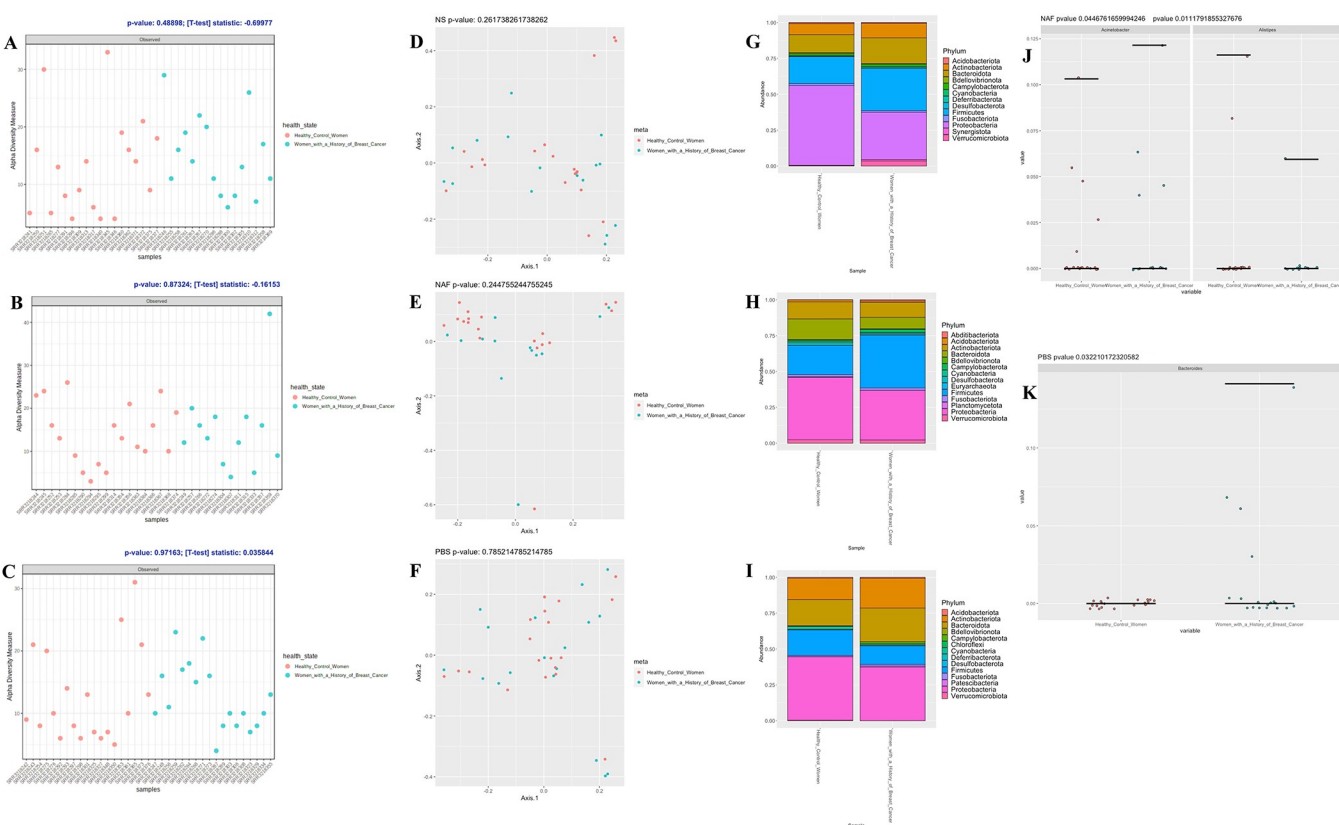

**Fig 3. Chan study re-analysis results.** (A) Nipple skin (NS) Observed OTU metric with p-value of 0.489. (B) Nipple aspirate fluid (NAF) Observed OTU metric with p-value of 0.873. (C) Post-betadine skin (PBS) Observed OTU metric with p-value of 0.972. (D-F) Comparing healthy control women and women with a history of breast cancer across (D) NS, (E) NAF, and (F) PBS samples. (G-I) Comparing proportional abundance of healthy control women and women with a history of breast cancer across (G) NS, (H) NAF, and (I) PBS samples. (J) NAF: Kruskal-Wallis test result of differentially abundant genera, *Acinetobacter* and *Alistipes*. (K) PBS: Kruskal-Wallis test result of differentially abundant genus *Bacteroides*.

In the beta diversity analysis, we have implemented a Bray-Curtis dissimilarity metric and performed PCoA using the rarefied ASV abundances as input where the genus-level ASVs were used in PCoA. The Adonis test was used to test for compositional differences, and the adonis function in the vegan package [19] from R was used to implement this test. The healthy control and cancer samples from the NS microbiota appear to separate into clusters, but when running the Adonis test there were no significant differences found in the bacterial composition with an unadjusted p-value of 0.262 (Fig 3D). The healthy control and cancer samples from the NAF microbiota appear to separate into clusters, but when running the Adonis test there were no significant differences found in the bacterial composition with an unadjusted p-value of 0.254 (Fig 3E). The healthy control and cancer samples from the PBS microbiota appear to separate into clusters, but when running the Adonis test there were no significant differences found in the bacterial composition with an unadjusted p-value of 0.785 (Fig 3F).

In the proportional abundance analysis, we have assessed the taxonomic composition of NS, NAF, and PBS microbiota at the phylum level. The proportional abundance plots include the top 300 sequences in order to clearly show the taxa present at the phylum level. The NS microbial composition was predominantly comprised of the phyla *Proteobacteria*, *Firmicutes*, and *Bacteroidota* (Fig 3G). The NAF microbial composition was predominantly comprised of the phyla *Proteobacteria* and *Firmicutes* (Fig 3H). The PBS microbial composition was predominantly comprised of the phyla *Proteobacteria*, *Firmicutes*, *Bacteroidota*, and *Actinobacteriota* (Fig 3I).

In the differential abundance analysis, the Kruskal-Wallis test was performed on the healthy and cancer samples from NS, NAF, and PBS ASVs and this test was performed through the kruskal.test function in the stats package [20] in R. The NS ASVs were not significantly different when comparing the healthy and cancer samples through the Kruskal-Wallis test. There were two NAF ASVs identified to be significantly different in relative abundance when comparing the healthy and cancer samples through the Kruskal-Wallis test, and they were *Alistipes* and *Acinetobacter* at the genus level (Fig 3J) present in both healthy and cancer NAF samples with an unadjusted p-value of 0.045 for *Acinetobacter* and 0.011 for *Alistipes*. There was one PBS ASV identified to be significantly different in relative abundance when comparing the healthy and cancer samples through the Kruskal-Wallis test, and it was *Bacteroides* at the genus level present in only cancer PBS samples with an unadjusted p-value of 0.032 (Fig 3K).

## Discussion

In each of the studies evaluated in this work, the microbial biomass being assessed is rather low as compared to, for instance, the microbial biomass from a gut microbiome assessment. Low biomass microbiome studies can be potentially confounded by environmental or methodological contamination. Each of the three studies attempted to control for this, however, the Chan et al. work was the only one to include actual environmental control samples in their collection and analysis workflow to positively identify and remove potential contaminants. The individual studies implemented tools, pipelines, and methods that are outdated and are not currently used for bioinformatic analyses, such as IM-TORNADO [21] and the Greengenes reference database [9]. While we have observed results that are similar to that of each original study's results such as when distinguishing between cancer and healthy samples, there are differences present. Our results have identified more taxa than the original studies and have shown that some taxa identified to be significant in the original paper are not found to be significant when re-analyzed with more recent and up-to-date techniques and methods. As mentioned in the introduction, differences in both the bioinformatic pipeline and reference database can contribute to differences in taxonomic assignment and downstream analyses

results. The methods implemented in our re-analysis are updated and known to perform better than their older counterparts. The original studies were published in 2016 and performed analyses using software that were available during that time. Specifically, the Hieken et al. study analyzed their 16S rRNA data using the IM-TORNADO [21] bioinformatics pipeline, the Urbaniak et al. study analyzed their 16S rRNA data using the Uclust algorithm of USEARCH version 7 [22], and the Chan et al. study analyzed their 16S rRNA data using mothur [23]. The studies have used software relevant to 2016 and since then these software have been updated, where now USEARCH has been updated to version 11, mothur has been updated to version 1.48.0, and the IM-TORNADO bioinformatics pipeline was deprecated in 2017. The software we used to analyze the data is most current and up-to-date, and provides new insight into the breast cancer microbiome. If we were to use the latest versions of the tools the original authors used, all but IM-TORNADO are several versions more advanced. Even so, there are clear reasons to prefer our ASV-based approach for these types of analyses as ASVs offer better resolution than OTUs, have better specificity, are independent of reference database, and have lower spurious sequence rates [24]. Additionally, the ASV table approach through DADA2 provides greater resolution and lowers false positives, and the SILVA database is known to perform better than the Greengenes database [25]. Primarily, SILVA is a more relevant and accurate database than Greengenes, as Greengenes has been deprecated and is not recommended for use since it was last updated August 2013. Additionally, continued use of Greengenes has been empirically shown to underperform relative to more updated databases [25].

The Hieken et al., Urbaniak et al., and Chan et al. studies are seminal papers in the breast cancer microbiome field, and these studies were not able to get a complete and accurate picture of the breast microbiome. Through our re-analysis, we were able to improve and expand upon the original results using modern best practices and were able to uncover new findings and correct unintentional errors in the original results. These findings are important as they will help to elucidate the direction that researchers interested in the breast cancer microbiome should move towards to investigate the appropriate taxa found in the breast microbiome. Therefore, re-analyses of past results and studies are needed to provide the most accurate and reliable results in the scientific community.

In the Hieken et al. study's re-analysis, our main findings were that breast and skin tissue microbiota showed no statistically significant differences for both the unweighted and weighted UniFrac distances, breast tissue adjacent to the cancer disease state and breast tissue adjacent to the benign disease state showed no statistically significant differences for the two UniFrac distances, the microbiota of the four communities (buccal swab, skin swab, skin tissue, and breast tissue) clustered separately from one another, and through a permutation test twelve differentially abundant taxa were identified as significant when comparing the benign and cancer disease states of breast tissue. Additionally, the proportional abundance analysis identified the following major phyla in similar abundances for the breast and skin tissue microbiota, *Actinobacteriota*, *Bacteroidota*, *Firmicutes*, and *Proteobacteria* where only the skin tissue had *Fusobacteriota* as a major phylum. In the original analysis, the Hieken et al. study's main findings were breast and skin tissue microbiota showed statistically significant differences for the unweighted UniFrac distance, breast tissue adjacent to the cancer disease state and breast tissue adjacent to the benign disease state showed statistically significant differences for the unweighted UniFrac distance, the microbiota of the four communities (buccal swab, skin swab, skin tissue, and breast tissue) clustered separately from one another, and through a permutation test five differentially abundant taxa were identified as significant when comparing the benign and cancer disease states of breast tissue. The Hieken et al. study's proportional abundance analysis identified the following major phyla in similar abundances for the breast and skin tissue microbiota, *Actinobacteria*, *Bacteroidetes*, *Firmicutes*, and *Proteobacteria*.

Comparing our re-analysis findings to the original study's results, the Hieken et al. study's analysis found differences to be significant for the unweighted UniFrac distance when comparing breast and skin tissue microbiota and when comparing adjacent breast tissue of cancer and benign disease states. In our re-analysis, these differences were not found to be significant for both UniFrac distances, and in the proportional abundance analysis our findings identified *Fusobacteriota* in skin tissue which was not identified in the original study's results. Additionally, our findings identified twelve differentially abundant taxa when comparing benign and cancer disease state breast tissue and the original study's result identified five differentially abundant taxa where one taxon was in common between our re-analysis and the original study's results, *Atopobium*. However, the abundance of *Atopobium* was switched as, in our re-analysis, it was found to be more abundant in the benign disease state whereas in the original results it was found to be more abundant in the cancer disease state.

In the Urbaniak et al. study's re-analysis, our main findings were that the breast tumor cancer samples cluster separately from the healthy samples with some overlap, *Pseudomonas* and *Bacillus* were abundant across all samples (benign, cancer, and healthy samples) where *Escherichia-Shigella* was prevalent in the cancer samples, and 18 statistically significant genera were identified of which *Bacillus* was in higher abundance in breast tumor cancer samples. In the original analysis, the Urbaniak et al. study's main findings were that the breast tumor cancer samples show a clear separation from the healthy samples cluster with no overlap, *Micrococcus* was in high abundance in healthy and benign samples with *Bacillus* being prevalent in breast tumor cancer samples, and ten statistically significant genera were identified of which *Bacillus*, *Staphylococcus*, *Enterobacteriaceae* (unclassified), *Comamondaceae* (unclassified), *and Bacteroidetes* (unclassified) showed significantly higher abundance in breast tumor cancer samples. Comparing the original results and our re-analysis results, there were many similarities between the two analyses results such as in both the original analysis and our re-analysis *Bacillus* was identified as statistically significant and was in higher abundance in breast tumor cancer samples and *Micrococcus* was identified as statistically significant and was in higher abundance in healthy samples. However, our re-analysis found the healthy samples and breast tumor cancer samples clustered separately with some overlap, identified 18 statistically significant genera, and showed a prevalent abundance of *Escherichia-Shigella* in the cancer samples in the proportional abundance analysis.

In the Chan et al. study's re-analysis, our main findings were that no significant differences were found in the beta diversity analysis when comparing healthy control and cancer samples for NS, NAF, and PBS microbiota, two NAF ASVs (*Acinetobacter* and *Alistipes*) and one PBS ASV (*Bacteroides*) were found to be significant in the differential abundance analysis, and in the proportional abundance the NS microbial composition was predominantly comprised of the phyla *Proteobacteria*, *Firmicutes*, and *Bacteroidota*, the NAF microbial composition was predominantly phyla *Proteobacteria* and *Firmicutes*, and the PBS microbial composition was predominantly the *Proteobacteria*, *Firmicutes*, *Bacteroidota*, and *Actinobacteriota* phyla. In the original analysis, the Chan et al. study's main findings were that there were no significant differences found when comparing healthy control and cancer samples for NS and PBS microbiota but was significant for the NAF microbiome, two NAF OTUs (*Alistipes* and *Sphingomonadaceae* (unclassified)) were found to be significant when comparing healthy control and cancer samples, and in the proportional abundance the NS microbial composition was predominantly comprised of the phyla *Bacteroidetes*, *Firmicutes*, *and Proteobacteria*, the NAF microbial composition was predominantly phyla *Firmicutes*, *Proteobacteria*, and *Bacteroidetes*, and the PBS microbial composition was predominantly the *Proteobacteria*, *Firmicutes*, *Bacteroidetes*, and *Actinobacteria*, *and Acidobacteria* phyla. Comparing the original results and our re-analysis results, there were similarities such as *Alistipes* was found to be significant in

the NAF microbiome in both the original analysis and re-analysis. There were also differences such as in our re-analysis, *Bacteroides* was identified as differentially abundant for PBS samples when comparing healthy control and cancer samples, but *Sphingomonadaceae* (unclassified) was not identified as differentially abundant.

The differences in the results between the original study's analysis and our re-analysis are potentially due to our use of substantially improved analytical pipelines and databases. Pipeline selection is known to have an effect on the resolution and identity of the taxa detected [24, 26, 27]. Some of this is in part due to the use of ASVs versus OTUs in the pipeline [24, 26], in other cases, published pipelines have retroactively been found to have biases [27]. Database selection is also known to dramatically influence taxonomic identification [28]. This should be unsurprising especially in the case of the Greengenes database as it has been more than a decade since it has been updated.

The results found from our re-analysis offer new insight into the breast tissue microbial community in healthy, benign, and malignant disease states across different variable regions and patient cohorts. The findings from our re-analysis are able to identify and define more taxa than what was previously reported in each study. Through our re-analysis, we identified more taxa because the taxonomic databases are more updated and, since the microbiome field is a rapidly evolving field, we have an exponentially better understanding of the human microbial composition. Additionally, with the availability of tools that provide improved resolution and a detailed look into 16S rRNA gene data, we identified more taxa than the original studies' results [24, 28]. There were some overlaps between the findings from our re-analysis, such as *Escherichia-Shigella* was found to be differentially abundant in invasive cancer breast tissue in the Hieken et al. re-analysis and it was also found to be prevalent in the cancer samples in the proportional abundance plot of the Urbaniak et al. re-analysis. Also, *Veillonella* was found to be differentially abundant in benign breast tissue in the Hieken et al. re- analysis and was also found to be differentially abundant in healthy breast tissue in the Urbaniak et al. re-analysis, and, similarly, *Acinetobacter* was found to be differentially abundant in the NAF cancer disease state in the Chan et al. re-analysis and in healthy breast tissue in the Urbaniak et al. re-analysis. There were similarities across the re-analysis results for each study, especially when the tissue and disease states were similar as shown between the Hieken et al. re-analysis and Urbaniak et al. re-analysis, and these similarities may shine light on the microbial communities that could be present in the breast microbiome across disease states. The different sample types and limited patient sample size among the studies contributed to each study having some of its own unique results with some overlaps as noted above. The samples from the Hieken et al. study and Urbaniak et al. study were similar enough that some similarities were found in the differential and proportional abundance results, as both studies primarily investigated breast tissue and compared across cancer and noncancer (benign or healthy) samples. However, the Chan et al. paper primarily studied the NAF and NS of the breast which was of lower biomass and a different microbial niche from the breast tissue which is why there was minimal overlap between the Chan et al. study and the other two studies. The cohorts used in these studies had limited patient numbers and it is possible that a future larger scale evaluation would further refine, update, or alter the findings of these pilot studies.

In our reevaluation of these three microbiome studies [2–4], we discovered that reanalyses are necessary for every study that studies the microbiome, especially older 16S studies. Reanalyses are important because they provide new insights to the microbiome field and help to assess robusticity of previously published findings by using new and updated tools and databases. In our reevaluation, there were false positives found from the original studies' results, as taxa that were originally identified to be significant were not found to be significant in our reanalyses. Additionally, new taxa were identified when assessing microbial abundance of

which were not reported in the original studies' findings, and multiple taxa were found to be significant between different disease states across the three studies that were not reported in the original studies' results. Through our reanalyses, we have deduced that every study that studies the microbiome, especially 16S studies, should be continuously reevaluated as the tools and databases develop.

## Materials and methods

### Dataset overview

The purpose of showing the specific alpha-diversity metricizes (or none) for each of the reanalyses is to do a like-for-like comparison between original and new analytical findings; hence, methods were applied based on the methods of the original studies.

The study *The Microbiome of Aseptically Collected Human Breast Tissue in Benign and Malignant Disease* by Hieken et al. in August 2016 [3], targets the V3-V5 variable region of their 16S rRNA data and compares the microbial communities between malignant and benign tumors of patients with breast cancer, using normal adjacent tissues as controls. Their data is comprised of 98 forward and 98 reverse fastq files with a patient sample size of 33 patients (Fig 4A), of which there were 16 women with benign disease and 17 women with malignant disease. The tissues of interest are buccal swab, skin swab, breast tissue, and breast skin tissue, which is also referred to as skin tissue in the paper. The sequencing data for this study is available from the NCBI SRA [29] and was downloaded from the SRA with accession number PRJNA335375, using the SRA explore interface at sra-explorer.info [30]. This study controlled for contaminants by implementing a sample collection protocol that minimized contamination during sample collection and sample storage. They analyzed their 16S rRNA paired-end data by implementing the IM-TORNADO bioinformatics pipeline [21] using the Greengenes

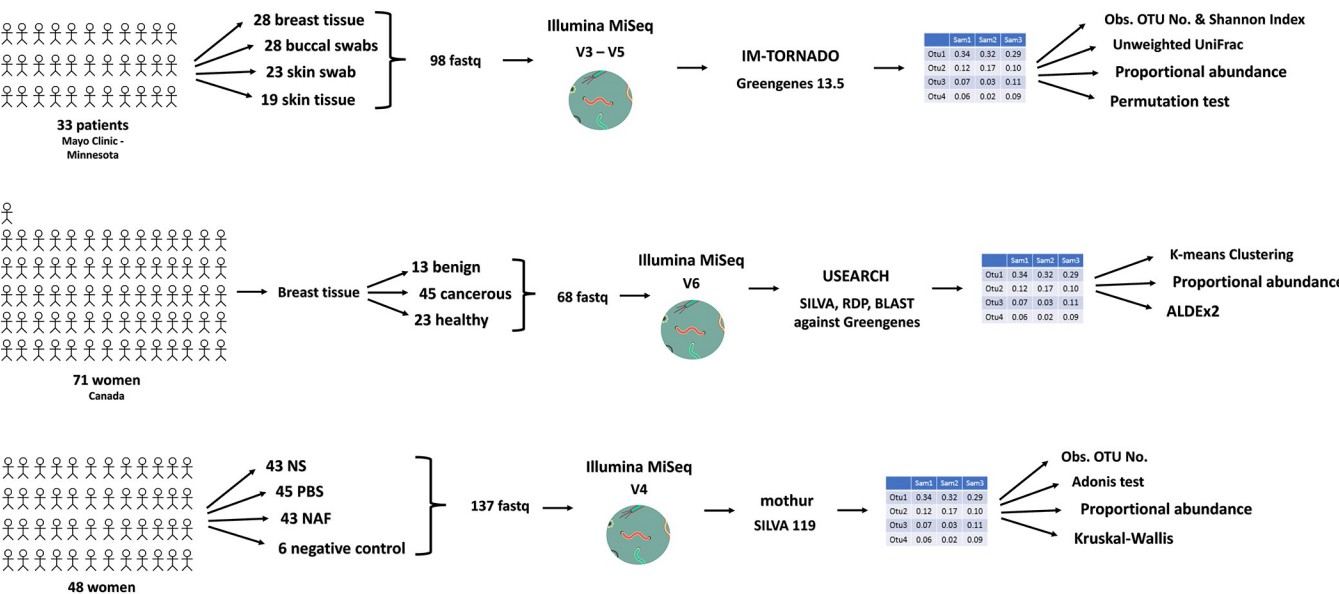

**Fig 4. Overview of the studies of interest.** (A) Hieken study overview outlining their patient sample size, number of samples, sequencing technology and variable region, bioinformatics pipeline and taxonomic reference database, and downstream analyses. (B) Urbaniak study overview outlining their patient sample size, number of samples, sequencing technology and variable region, bioinformatics pipeline and taxonomic reference database, and downstream analyses. (C) Chan study overview outlining their patient sample size, number of samples, sequencing technology and variable region, bioinformatics pipeline and taxonomic reference database, and downstream analyses.

version 13.5 reference database [9]. Furthermore, they implemented two alpha diversity measures which were the observed OTU number and the Shannon Index, two beta diversity measures which were the unweighted and weighted UniFrac distances [13], and performed differential abundance analysis to identify significant taxa. The Hieken et al. study did not perform multiple correction and reported unadjusted p-values for their statistical data analyses.

The study *The Microbiota of Breast Tissue and Its Association with Breast Cancer* by Urbaniak et al. in August 2016 [4], targets the V6 variable region of their 16S rRNA data and compares the microbial composition between breast tissue from healthy control women and normal adjacent breast tissue from women with breast cancer, and they also compare normal adjacent breast tissue and breast tumor tissue. Their data is comprised of 68 merged fastq files with a patient sample size of 71 patients (Fig 4B), of which there were 13 benign, 45 cancerous, and 23 healthy samples. The samples of interest in this study are malignant, benign, and healthy samples from breast tissue in this paper, where the samples are normal adjacent tissue from women with breast cancer which is either benign or malignant and tissue from healthy controls. The sequencing data for this study is available from the NCBI SRA [29] and was downloaded from the SRA with accession number SRP076038, using the SRA explore interface at sra-explorer.info [30]. The singleton OTUs were removed and OTUs were kept if they represented $\geq$ 2% of reads in at least one sample. They analyzed their 16S rRNA paired-end data by clustering the reads into OTUs by 97% identity using the Uclust algorithm within USEARCH version 7 [22]. Taxonomic assignment of each OTU was made by extracting best hits from SILVA database [11], manually verifying using RDP SeqMatch tool [10], and then using BLAST [31] against the Greengenes database [9], where the hits with the highest percent identity and coverage were used to assign taxonomy. The OTU sequences were aligned using MUSCLE [32] and were inputted to FASTTREE [33] to build a tree of OTU sequences where PCoA plots of weighted UniFrac distances [8] were made in QIIME [34] using this tree of OTU sequences.

Additionally, unsupervised k-means clustering was performed using euclidean distances of center-log ratio (CLR) transformed data and used the ALDEx2 R package [18] to measure the relative abundances of genera where Benjamini-Hochberg corrected p-value of the Wilcoxon rank test was used to test for significance. Furthermore, to test for differences between microbiota a microbiome regression-based kernel association test (mirkat) was performed in R using the MiRKAT package [35] where a kernel metric was built using UniFrac distances and Bray-Curtis dissimilarity metric. UniFrac distances and Bray-Curtis dissimilarity metric were both used in the kernel metric simultaneously.

The study *Characterization of the microbiome of nipple aspirate fluid of breast cancer survivors* by Chan et al. in June 2016 [2], targets the V4 variable region of their 16S rRNA data and investigates the microbial community present in NAF and their potential association with breast cancer by comparing NAF between healthy control samples and women with history of breast cancer samples. Their data is comprised of 137 forward and 137 reverse fastq files with a patient sample size of 48 patients (Fig 4C), of which there were 23 healthy control women and 25 women with a history of cancer samples. There are healthy control and women with a history of breast cancer samples and tissues of interest are NAF, NS, and PBS in this paper. The women with a history of breast cancer are also referred to as cancer samples in this section. The sequencing data for this study is available from the NCBI SRA [29] and was downloaded from the SRA with accession number PRJNA314877, using the SRA explore interface at sra-explorer.info [30]. There were OTUs detected in empty control Eppendorf tubes and were removed from analyses to account for contaminating microbial 16S rDNA sequences from extraction kits and reagents. They analyzed their 16S rRNA paired-end data by implementing the Schloss MiSeq standard operating procedure in mothur [23] using the SILVA version 119

reference database [11]. They implemented an alpha diversity measure which was the observed OTU number, a beta diversity measure which was the analysis of variance (Adonis) test using Bray-Curtis distance and PCoA, and performed differential abundance analysis through the Kruskal-Wallis test to identify significant taxa. The Adonis test was used to measure community composition differences and a nonparametric Kruskal-Wallis test was used to test whether the OTU relative abundances were statistically significant between the healthy control and breast cancer samples for NAF, NS, and PBS samples. The OTU table was rarefied for calculating dissimilarity measures and PCoA to account for any bias from uneven sequencing depth. The statistical analyses used a p-value cutoff of 0.05 and false discovery correction was not applied when comparing the OTU relative abundances; however, when performing functional prediction with Phylogenetic Investigation of Communities by Reconstruction of Unobserved States (PICRUSt) [36] and Kyoto Encyclopedia of Genes and Genomes (KEGG) [37] a multiple test correction was applied.

## Preprocessing

In the Hieken and Chan studies, prior to beginning denoising, the primers used to amplify different regions of the 16S rRNA gene were trimmed from the unzipped paired end fastq files using cutadapt version 1.15 [38]. In the Urbaniak study, upon downloading the fastq files from the SRA [29], the files were already merged and pre-processed; hence, primers were not trimmed.

The Hieken study targeted the V3-V5 regions, the Chan study targeted the V4 region, and the Urbaniak study targeted the V6 region of the 16S rRNA gene. In the Hieken study, the 357F primer used was `AATGATACGGCGACCACCGAGATCTACACTATGGTAATTGTCCTA CGGGAGGCAGCAG` and the 926R primer used was `CAAGCAGAAGACGGCATACGAGATGCCG CATTCGATXXXXXXXXXXXXXCCGTCAATTCMTTTRAGT`. In the Chan study, the F515 primer used was `AATGATACGGCGACCACCGAGACGTACGTACGGTGTGCCAGCMGCCGCGGTAA` and the R806 primer used was `CAAGCAGAAGACGGCATACGAGATXXXXXXXXXXXXXACGTACG TACCGGATACHVGGGTWTCTAAT`. In the Urbaniak study, the V6-forward primer used was 5′`ACACTCTTTCCCTACACGACGCTCTTCCGATCTnnnn(8)CWACGCGARGAACCTTACC`3′ and the V6-reverse primer used was 5′`CGGTCTCGGCATTCCTGCTGAACCGCTCTTCCGATC Tnnnn(8)ACRACACGAGCTGACGAC`3′.

After trimming the primers, FastQC version 0.11.5 [39] was used to assess the Phred quality score of each forward and reverse fastq file as well as to check for potential artifactual sequences in the dataset. In the Urbaniak study, the files were already merged and pre-processed; hence, FastQC was not used to assess the quality scores.

## DADA2 pipeline analysis

The trimmed reverse fastq files from the Hieken and Chan data were of very low quality and few of them made it through the quality control steps within the pipeline; therefore, only forward fastq files were included in the analysis, as these reads met our quality control standards. The trimmed forward fastq files from the Hieken and Chan data and merged fastq files from the Urbaniak data were analyzed using default parameters in the DADA2 1.16 pipeline [40], with changes in the filterAndTrim command, makeSequenceTable command, and the mergers command. In the filterAndTrim command, the reverse file input and truncLen parameter were removed and the maxEE value was adjusted from the default maxEE = c(2,2) to maxEE = 3. These changes were implemented to allow for flexibility and to relax the stringency in our analysis so to prevent massive removal of sample reads which would make interpretation of the results impossible. Also, the multithread parameter was set to equal two to avoid

memory issues. In the makeSequenceTable command, the dadaFs variable was used to make the seqtab variable instead of the default mergers variable. The mergers command was removed as the analysis was performed with only the forward fastq files as input. Additionally, the SILVA version 138 reference database [11] compatible with DADA2 [26] was downloaded from the Zenodo site [41]. The ASVs detected in control samples were removed from analyses to account for contaminants for the Chan et al. study. The tax_glom and transform_sample_counts functions in phyloseq were implemented to make the proportional abundance plots. Additionally, the microshades package was used to make colorblind friendly proportional abundance plots and the code can be found at the microshades website [42]. In order to make the microshades plot, the phyloseq object is used as input to the prep_mdf function. After making the proportional abundance plots, phylogenetic analyses were performed.

## Phylogenetic analysis

The phylogenetic code was adapted from the compbiocore github site [43], specifically commands from sequences<-getSequences(seqtab.nochim) up to dm <- dist.ml(phang.align) were implemented. The tree was generated using dm as input to the upgma command from the phangorn R package [44]. The amplicon sequence variant (ASV) table and taxonomy table outputs from DADA2 along with the metadata and phylogenetic tree files were used as input to the MicrobiomeAnalyst web interface [45] for visual exploratory data analysis.

## Statistical analysis

In order to calculate the UniFrac distances [13] for the Hieken and Urbaniak data, the GUniFrac function from the GUniFrac R package [12] was implemented where the input was the ASV table output from the DADA2 analyses. The UniFrac distances were used to perform beta diversity analyses for the Hieken and Urbaniak data. Additionally, to perform statistical analyses for the Hieken data, a 10% prevalence filter and total sum scaling was applied to the ASV table to identify differentially abundant taxa. A permutation test was also performed for the Hieken data to identify significant ASVs of which were visualized through bar plots. In order to perform statistical analyses for the Urbaniak data, both UniFrac and Bray-Curtis distances were used to perform beta diversity analyses and euclidean distances were used to perform K-Means Clustering. Additionally, for the Urbaniak data, the ALDEx2 R package [18] was used to measure the relative abundances of statistically significant genera. The ASVs in the ALDEx2 output were used to link associated taxa at the genus level to the ALDEx2 output and each genus in the output was visualized through box plots.

In order to perform statistical analyses for the Chan data, the Adonis test and Kruskal-Wallis test were run. The Adonis test was run to measure community composition differences and a nonparametric Kruskal-Wallis test was run to test whether relative abundances of ASVs were statistically significant between the healthy control and breast cancer samples for NAF, NS, and PBS samples. The p-values reported for the Adonis and Kruskal-Wallis tests are unadjusted p-values as the Chan study also reported unadjusted p-values. The Bray-Curtis distance and a proportionally rarefied phyloseq object were used to perform the Adonis test. The Kruskal-Wallis test was then run for each ASV from a rarefied phyloseq object, and ASVs that are not significant are filtered out. The significant ASVs were then linked with their associated taxa and the abundance values for each taxon were visualized through a dot plot.

## Acknowledgments

Thank you to the Hieken et al., Urbaniak et al., and Chan et al. studies for making their data publicly available.

## Author Contributions

**Conceptualization:** Sidra Sohail, Michael B. Burns.

**Formal analysis:** Sidra Sohail.

**Investigation:** Sidra Sohail.

**Methodology:** Sidra Sohail, Michael B. Burns.

**Project administration:** Michael B. Burns.

**Supervision:** Michael B. Burns.

**Validation:** Michael B. Burns.

**Visualization:** Sidra Sohail, Michael B. Burns.

**Writing – original draft:** Sidra Sohail, Michael B. Burns.

**Writing – review & editing:** Sidra Sohail, Michael B. Burns.

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
