## [Decision Letter · Decision Letter 0]

28 Jun 2023

PONE-D-23-13336Integrating current analyses of the breast cancer microbiomePLOS ONE

Dear Dr. Sohail,

Thank you for submitting your manuscript to PLOS ONE. After careful consideration, we feel that it has merit but does not fully meet PLOS ONE’s publication criteria as it currently stands. Therefore, we invite you to submit a revised version of the manuscript that addresses the points raised during the review process.

We look forward to receiving your revised manuscript.

Kind regards,

Rajesh P. Shastry

Academic Editor

PLOS ONE

Reviewers' comments:

Reviewer's Responses to Questions

**Comments to the Author**

1. Is the manuscript technically sound, and do the data support the conclusions?

Reviewer #1: Yes

Reviewer #2: Partly

2. Has the statistical analysis been performed appropriately and rigorously? 

Reviewer #1: Yes

Reviewer #2: Yes

3. Have the authors made all data underlying the findings in their manuscript fully available?

Reviewer #1: Yes

Reviewer #2: Yes

4. Is the manuscript presented in an intelligible fashion and written in standard English?

Reviewer #1: Yes

Reviewer #2: Yes

5. Review Comments to the Author

Reviewer #1: In this manuscript, authors reanalied the published data on breast cancer and microbiome. The reanalyses are important because they provide new insights to the microbiome field and help to assess robusticity of previously published findings by using new and updated tools and methods with the updated databases. It provide useful information on the breast cancer and microbiome with newly identified significant genera by kicking off false positives.

The manuscript should be reorganized.

Reviewer #2: I would like to thank the authors for the opportunity to review this analysis. It is an interesting field with indeed a lot of methodological considerations.

Please see below my comments:

1. It is difficult to understand the scope of the manuscript, other than to explore whether novel methods can confirm previously reported results. This should be clearer in the introduction-

2. Some general comments are the lack of explaining the abbreviations in the figure legends and when they first appear in the manuscript. Also, in some places the wording is difficult to understand and there is inconsistency in the tense used, since present and past tense are used in different places when, for example, presenting the results. I would urge the authors to carefully go through the manuscript and look at the abbreviations and improve clarity of the text.

3. The studies are presented in different order in the results part and the methods part which can cause confusion. Please be consistent in presenting every time in the same order.

4. Lines 77-79, I would suggest removing bold from this sentence since it does provide any functionality.

5. Lines 87-89, an example of text that should be grammatically revised.

6. Different methods have been applied to different cohorts. For example, alpha-diversity was presented as both OTU numbers and Shannon Index in the Hieken cohort, whereas no Shannon Index was presented in the Chan cohort and no alpha-diversity analysis is presented for the Urbaniak cohort. If this is due to methodological considerations, it should be clearly stated.

7. In lines 142-145, it is stated “In the proportional abundance analysis, we have assessed the taxonomic composition of breast and skin tissue microbiota, breast tissue microbiota in benign, also known as BBD_non_atypia, and invasive cancer, also known as InvCa, disease states, and buccal and skin swab microbiota at phylum, family, and genus levels.” These are not known as such; they are presented in the manuscript figures as such. Please correct accordingly.

8. Please provide rationale or reference on the choice of the 10% prevalence and 0,2% relative abundance cut-offs in the differential abundance analysis of the Hieken cohort (lines 158-159).

9. There is no mention in the methods on the aspects of contaminants. How were these handled? What bioinformatics methods were used? Did your methods identify differences compared to previous analyses?

10. Although the articles are referred to in the introduction, they should still be referenced at the begin of the results and the methods presentation, when each cohort is being introduced.

11. Please provide more reference on your statement “The methods implemented in our re-analysis are updated and known to perform better than their older counterparts” (lines 259-260).

12. Lines 261-262, in which regard is SILVA database better than Greengenes database. Please elaborate more in the text.

13. Line 265-267: they is currently not best practice and I believe it is an exaggeration to call the results a new phenomenon. Newer techniques have been shown to improve analytical methods in other fields as well, but this should not prompt previous publications to be called mistaken. I would therefore urge the authors to rephrase this sentence.

14. Figure 4: excellent figure to give an overview of the different cohorts. However, not clearly indicated what type of samples are included in the study by Chan et al (4C), especially since this is very different than the other two.

15. The Ethics statement includes repetition of more or less the same statement in different sentences. Please revise and provide a more consist statement without repetitions. Also, if you call the data “ethically” collected, you should provide reference and rationale which to call it such. I would urge the authors to avoid using this expression.

16. Lines 328-332 should be part of the introduction.

17. In the discussion I would suggest providing more information on the initial findings, compare them at study level with the current results and discuss on what you believe these differences could depend on.

18. In your opinion, why do you think your results identify more taxa? Please elaborate in the discussion.

19. Please also discuss the different type of samples in each cohort and if you encounter any challenges between these different samples.

20. These are cohort with small number of patients and small changes could have significant impact on the results. Please address this in your discussion.

21. The quality of the graphs is bad, but I am not sure if this is due to the journal, provided that this is review material. If this is the resolution of the original images, then they should be improved prior to potential publication.

6. PLOS authors have the option to publish the peer review history of their article (what does this mean?). If published, this will include your full peer review and any attached files.

Reviewer #1: No

Reviewer #2: No

---

## [Author Response · Author response to Decision Letter 0]

11 Aug 2023

Response to Reviewers

Reviewer 1: “In this manuscript, authors reanalied the published data on breast cancer and microbiome. The reanalyses are important because they provide new insights to the microbiome field and help to assess robusticity of previously published findings by using new and updated tools and methods with the updated databases. It provide useful information on the breast cancer and microbiome with newly identified significant genera by kicking off false positives.

The manuscript should be reorganized.”

Response: Thank you for the feedback. We are glad that you agree that these are important findings to publish. As per your suggestion, the manuscript has been reworked with additional clarifying texts please see comments for Reviewer 2.

Reviewer 2: “I would like to thank the authors for the opportunity to review this analysis. It is an interesting field with indeed a lot of methodological considerations. Please see below my comments:”

Reviewer 2, Point 1: “It is difficult to understand the scope of the manuscript, other than to explore whether novel methods can confirm previously reported results. This should be clearer in the introduction-”

Response: Thank you for the feedback. We agree that we can make the scope of the project more clear and in order to do so we have taken your suggestion and added additional clarifying text to the first paragraph in the introduction to say this, “The microbiome field is rapidly changing as there are different variable regions, different databases, and different pipelines available, and with this variability it is important to perform retrospective analyses to assess reproducibility of the original studies’ results and to report any new findings from the updated analyses. Therefore, in this work, we are performing a retrospective analysis of existing studies on the breast cancer microbiome. Through the retrospective analysis, we report microbial taxa that were not initially identified as important by the original studies and have assessed the reproducibility of the original studies’ results.” and this provides a clearer scope and the context for this work. Additionally, we have added additional clarifying text to the third paragraph in the introduction to say this “Through our retrospective analysis of existing studies on the breast cancer microbiome, we report new findings on microbial taxa in association with the breast cancer microbiome that were initially not discovered by the original authors.” and this provides a clearer outline for this work as well. 

Reviewer 2, Point 2: “Some general comments are the lack of explaining the abbreviations in the figure legends and when they first appear in the manuscript. Also, in some places the wording is difficult to understand and there is inconsistency in the tense used, since present and past tense are used in different places when, for example, presenting the results. I would urge the authors to carefully go through the manuscript and look at the abbreviations and improve clarity of the text.”

Response: Thank you for this feedback. We noted several different instances that meet this criterion and we have gone through and fixed those.

Reviewer 2, point 3: “The studies are presented in different order in the results part and the methods part which can cause confusion. Please be consistent in presenting every time in the same order.”

Response: Thank you, we have rearranged the results section, so that it aligns with how the studies are presented in the materials and methods section.

Reviewer 2, Point 4: “Lines 77-79, I would suggest removing bold from this sentence since it does provide any functionality.”

Response: Thank you for the feedback. We have removed the bold format from the sentence.

Reviewer 2, Point 5: “Lines 87-89, an example of text that should be grammatically revised.”

Response: Thank you, we appreciate your feedback and we have revised this sentence, “As the field of microbiome research advances, the reference databases are routinely being updated and massively expanded and corrected. While the Greengenes database [9] was the best tool at the time, it is now deprecated as it has not been updated for a decade. Modern best-practices rely on up-to-date databases, including the Ribosomal Database Project (RDP) [10] and the SILVA database [11].”

Reviewer 2, Point 6: “Different methods have been applied to different cohorts. For example, alpha-diversity was presented as both OTU numbers and Shannon Index in the Hieken cohort, whereas no Shannon Index was presented in the Chan cohort and no alpha-diversity analysis is presented for the Urbaniak cohort. If this is due to methodological considerations, it should be clearly stated.”

Response: Thank you for the feedback, and yes, this is something that we agree we should comment on. We have added the following clarifying text to the manuscript to make clear why this is the case in the Dataset Overview subsection of the Materials and Methods section, “The purpose of showing the specific alpha-diversity metricizes (or none) for each of the reanalyses is to do a like-for-like comparison between original and new analytical findings; hence, methods were applied based on the methods of the original studies.”

Reviewer 2, Point 7: “In lines 142-145, it is stated “In the proportional abundance analysis, we have assessed the taxonomic composition of breast and skin tissue microbiota, breast tissue microbiota in benign, also known as BBD_non_atypia, and invasive cancer, also known as InvCa, disease states, and buccal and skin swab microbiota at phylum, family, and genus levels.” These are not known as such; they are presented in the manuscript figures as such. Please correct accordingly.”

Response: Thank you for the feedback. We have included the following text to clarify the disease states nomenclature, “In the Hieken et al. study, the benign disease state is referred to as BBD_non_atypia and this stands for benign breast disease without atypia, and the cancer disease state is referred to as InvCa and this stands for invasive cancer. The unweighted UniFrac distance plot (Fig 1F) and taxonomic plot (Fig 1I) use the BBD_non_atypia and InvCa abbreviations as these were extracted from the Hieken et al. study’s publicly available metadata.”

Reviewer 2, Point 8: “Please provide rationale or reference on the choice of the 10% prevalence and 0,2% relative abundance cut-offs in the differential abundance analysis of the Hieken cohort (lines 158-159).”

Response: Thank you for the feedback. We have included the following text in the Hieken et al. Re-analysis Results subsection to provide rationale on our choice of the 10% prevalence and 0.2% relative abundance cut-offs, “In the differential abundance analysis, taxa with prevalence of less than 10% and relative abundance of less than 0.2% were filtered out, which are the same cut-offs as the Hieken et al. study implemented for their differential abundance analysis. Using these same thresholds allows for a like-for-like comparison between the original and updated findings.”

Reviewer 2, Point 9: “There is no mention in the methods on the aspects of contaminants. How were these handled? What bioinformatics methods were used? Did your methods identify differences compared to previous analyses?”

Response: Thank you for the feedback. We agree that removing contaminants are an important step for data analysis and have added the protocol that the original studies implemented in the dataset overview subsection of the Materials and Methods section. We have included the following text for the Hieken et al. study’s protocol “This study controlled for contaminants by implementing a sample collection protocol that minimized contamination during sample collection and sample storage.”, text for the Urbaniak et al. study’s protocol “The singleton OTUs were removed and OTUs were kept if they represented > 2% of reads in at least one sample.”, and text for the Chan et al. study’s protocol “There were OTUs detected in empty control Eppendorf tubes and were removed from analyses to account for contaminating microbial 16S rDNA sequences from extraction kits and reagents.” Additionally, we have described our contaminant removal protocol we implemented for the Chan et al. study and have included the following text in the DADA2 Pipeline Analysis subdivision of the Materials and Methods section “The ASVs detected in control samples were removed from analyses to account for contaminants for the Chan et al. study.” We implemented a contaminant removal protocol similar to the Chan et al. study’s contaminant protocol. Finally, we have also added comment text in the discussion section to briefly note that among these three studies, the Chan et al. work did the most thorough job of directly addressing and controlling for potential contaminants: “In each of the studies evaluated in this work, the microbial biomass being assessed is rather low as compared to, for instance, the microbial biomass from a gut microbiome assessment. Low biomass microbiome studies can be potentially confounded by environmental or methodological contamination. Each of the three studies attempted to control for this, however, the Chan et al. work was the only one to include actual environmental control samples in their collection and analysis workflow to positively identify and remove potential contaminants.”

Reviewer 2, Point 10: “Although the articles are referred to in the introduction, they should still be referenced at the begin of the results and the methods presentation, when each cohort is being introduced.”

Response: Thank you for the feedback, we have referenced the Hieken et al., Urbaniak et al., and Chan et al. studies in the beginning of the results section and methods section.

Reviewer 2, point 11: “Please provide more reference on your statement “The methods implemented in our re-analysis are updated and known to perform better than their older counterparts” (lines 259-260).”

Response: We thank you and appreciate your feedback, and we have added the following text to provide more reference. “The original studies were published in 2016 and performed analyses using software that were available during that time. Specifically, the Hieken et al. study analyzed their 16S rRNA data using the IM-TORNADO [21] bioinformatics pipeline, the Urbaniak et al. study analyzed their 16S rRNA data using the Uclust algorithm of USEARCH version 7 [22], and the Chan et al. study analyzed their 16S rRNA data using mothur [23]. The studies have used software relevant to 2016 and since then these software have been updated, where now USEARCH has been updated to version 11, mothur has been updated to version 1.48.0, and the IM-TORNADO bioinformatics pipeline was deprecated in 2017. The software we used to analyze the data is most current and up-to-date, and provides new insight into the breast cancer microbiome. If we were to use the latest versions of the tools the original authors used, all but IM-TORNADO are several versions more advanced. Even so, there are clear reasons to prefer our ASV-based approach for these types of analyses as ASVs offer better resolution than OTUs, have better specificity, are independent of reference database, and have lower spurious sequence rates [24].”

Reviewer 2, point 12: “Lines 261-262, in which regard is SILVA database better than Greengenes database. Please elaborate more in the text.”

Response: Thank you for your feedback, we have added the following text. “Primarily, SILVA is a more relevant and accurate database than Greengenes, as Greengenes has been deprecated and is not recommended for use since it was last updated August 2013. Additionally, continued use of Greengenes has been empirically shown to underperform relative to more updated databases [25].” 

Reviewer 2, point 13: “Line 265-267: they is currently not best practice and I believe it is an exaggeration to call the results a new phenomenon. Newer techniques have been shown to improve analytical methods in other fields as well, but this should not prompt previous publications to be called mistaken. I would therefore urge the authors to rephrase this sentence.”

Response: Thank you for the feedback. We have revised that sentence to temper our findings and say that “Through our re-analysis, we were able to improve and expand upon the original results using modern best practices and were able to uncover new findings and correct unintentional errors in the original results.”

Reviewer 2, point 14: “Figure 4: excellent figure to give an overview of the different cohorts. However, not clearly indicated what type of samples are included in the study by Chan et al (4C), especially since this is very different than the other two.”

Response: Thank you and we appreciate your feedback. We have revised figure 4 to show the sample types of all three studies.

Reviewer 2, point 15: “The Ethics statement includes repetition of more or less the same statement in different sentences. Please revise and provide a more consist statement without repetitions. Also, if you call the data “ethically” collected, you should provide reference and rationale which to call it such. I would urge the authors to avoid using this expression.”

Response: Thank you for the feedback and we have revised the Ethics Statement in the submission portal to “This manuscript is a re-analysis of previously collected publicly available datasets.”

Reviewer 2, point 16: “Lines 328-332 should be part of the introduction.”

Response: We thank you and appreciate your feedback, and we have included the following text to the introduction “Each study used a variety of different methods for their 16S rRNA gene data analysis. The beta diversity measures were calculated using an OTU table and phylogenetic tree. The observed OTU number reflects the species richness and the Shannon index reflects the species evenness. The unweighted UniFrac [8] measures differences in community presence such as whether or not there is an OTU present, and the weighted UniFrac measures differences in community abundance.”

Reviewer 2, point 17: “In the discussion I would suggest providing more information on the initial findings, compare them at study level with the current results and discuss on what you believe these differences could depend on.”

Response: Thank you, we have updated the discussion section to include the following text, “In the Hieken et al. study’s re-analysis, our main findings were that breast and skin tissue microbiota showed no statistically significant differences for both the unweighted and weighted UniFrac distances, breast tissue adjacent to the cancer disease state and breast tissue adjacent to the benign disease state showed no statistically significant differences for the two UniFrac distances, the microbiota of the four communities (buccal swab, skin swab, skin tissue, and breast tissue) clustered separately from one another, and through a permutation test twelve differentially abundant taxa were identified as significant when comparing the benign and cancer disease states of breast tissue. Additionally, the proportional abundance analysis identified the following major phyla in similar abundances for the breast and skin tissue microbiota, Actinobacteriota, Bacteroidota, Firmicutes, and Proteobacteria where only the skin tissue had Fusobacteriota as a major phylum. In the original analysis, the Hieken et al. study’s main findings were breast and skin tissue microbiota showed statistically significant differences for the unweighted UniFrac distance, breast tissue adjacent to the cancer disease state and breast tissue adjacent to the benign disease state showed statistically significant differences for the unweighted UniFrac distance, the microbiota of the four communities (buccal swab, skin swab, skin tissue, and breast tissue) clustered separately from one another, and through a permutation test five differentially abundant taxa were identified as significant when comparing the benign and cancer disease states of breast tissue. The Hieken et al. study’s proportional abundance analysis identified the following major phyla in similar abundances for the breast and skin tissue microbiota, Actinobacteria, Bacteroidetes, Firmicutes, and Proteobacteria. Comparing our re-analysis findings to the original study’s results, the Hieken et al. study’s analysis found differences to be significant for the unweighted UniFrac distance when comparing breast and skin tissue microbiota and when comparing adjacent breast tissue of cancer and benign disease states. In our re-analysis, these differences were not found to be significant for both UniFrac distances, and in the proportional abundance analysis our findings identified Fusobacteriota in skin tissue which was not identified in the original study’s results. Additionally, our findings identified twelve differentially abundant taxa when comparing benign and cancer disease state breast tissue and the original study’s result identified five differentially abundant taxa where one taxon was in common between our re-analysis and the original study’s results, Atopobium. However, the abundance of Atopobium was switched as, in our re-analysis, it was found to be more abundant in the benign disease state whereas in the original results it was found to be more abundant in the cancer disease state.

 In the Urbaniak et al. study’s re-analysis, our main findings were that the breast tumor cancer samples cluster separately from the healthy samples with some overlap, Pseudomonas and Bacillus were abundant across all samples (benign, cancer, and healthy samples) where Escherichia-Shigella was prevalent in the cancer samples, and 18 statistically significant genera were identified of which Bacillus was in higher abundance in breast tumor cancer samples. In the original analysis, the Urbaniak et al. study’s main findings were that the breast tumor cancer samples show a clear separation from the healthy samples cluster with no overlap, Micrococcus was in high abundance in healthy and benign samples with Bacillus being prevalent in breast tumor cancer samples, and ten statistically significant genera were identified of which Bacillus, Staphylococcus, Enterobacteriaceae (unclassified), Comamondaceae (unclassified), and Bacteroidetes (unclassified) showed significantly higher abundance in breast tumor cancer samples. Comparing the original results and our re-analysis results, there were many similarities between the two analyses results such as in both the original analysis and our re-analysis Bacillus was identified as statistically significant and was in higher abundance in breast tumor cancer samples and Micrococcus was identified as statistically significant and was in higher abundance in healthy samples. However, our re-analysis found the healthy samples and breast tumor cancer samples clustered separately with some overlap, identified 18 statistically significant genera, and showed a prevalent abundance of Escherichia-Shigella in the cancer samples in the proportional abundance analysis.

 In the Chan et al. study’s re-analysis, our main findings were that no significant differences were found in the beta diversity analysis when comparing healthy control and cancer samples for NS, NAF, and PBS microbiota, two NAF ASVs (Acinetobacter and Alistipes) and one PBS ASV (Bacteroides) were found to be significant in the differential abundance analysis, and in the proportional abundance the NS microbial composition was predominantly comprised of the phyla Proteobacteria, Firmicutes, and Bacteroidota, the NAF microbial composition was predominantly phyla Proteobacteria and Firmicutes, and the PBS microbial composition was predominantly the Proteobacteria, Firmicutes, Bacteroidota, and Actinobacteriota phyla. In the original analysis, the Chan et al. study’s main findings were that there were no significant differences found when comparing healthy control and cancer samples for NS and PBS microbiota but was significant for the NAF microbiome, two NAF OTUs (Alistipes and Sphingomonadaceae (unclassified)) were found to be significant when comparing healthy control and cancer samples, and in the proportional abundance the NS microbial composition was predominantly comprised of the phyla Bacteroidetes, Firmicutes, and Proteobacteria, the NAF microbial composition was predominantly phyla Firmicutes, Proteobacteria, and Bacteroidetes, and the PBS microbial composition was predominantly the Proteobacteria, Firmicutes, Bacteroidetes, and Actinobacteria, and Acidobacteria phyla. Comparing the original results and our re-analysis results, there were similarities such as Alistipes was found to be significant in the NAF microbiome in both the original analysis and re-analysis. There were also differences such as in our re-analysis, Bacteroides was identified as differentially abundant for PBS samples when comparing healthy control and cancer samples, but Sphingomonadaceae (unclassified) was not identified as differentially abundant.

 The differences in the results between the original study’s analysis and our re-analysis are potentially due to our use of substantially improved analytical pipelines and databases. Pipeline selection is known to have an effect on the resolution and identity of the taxa detected [24,26,27]. Some of this is in part due to the use of ASVs versus OTUs in the pipeline [24,28], in other cases, published pipelines have retroactively been found to have biases [27]. Database selection is also known to dramatically influence taxonomic identification [29]. This should be unsurprising especially in the case of the Greengenes database as it has been more than a decade since it has been updated.”

Reviewer 2, point 18: “In your opinion, why do you think your results identify more taxa? Please elaborate in the discussion.”

Response: Thank you for your question. We have added the following text to the discussion section, “Through our re-analysis, we identified more taxa because the taxonomic databases are more updated and, since the microbiome field is a rapidly evolving field, we have an exponentially better understanding of the human microbial composition. Additionally, with the availability of tools that provide improved resolution and a detailed look into 16S rRNA gene data, we identified more taxa than the original studies’ results [24,29].”

Reviewer 2, point 19: “Please also discuss the different type of samples in each cohort and if you encounter any challenges between these different samples.”

Response: Thank you, we appreciate your feedback and have the following text to the discussion,“The different sample types and limited patient sample size among the studies contributed to each study having some of its own unique results with some overlaps as noted above. The samples from the Hieken et al. study and Urbaniak et al. study were similar enough that some similarities were found in the differential and proportional abundance results, as both studies primarily investigated breast tissue and compared across cancer and noncancer (benign or healthy) samples. However, the Chan et al. paper primarily studied the NAF and NS of the breast which was of lower biomass and a different microbial niche from the breast tissue which is why there was minimal overlap between the Chan et al. study and the other two studies.”

Reviewer 2, point 20: “These are cohort with small number of patients and small changes could have significant impact on the results. Please address this in your discussion.”

Response: Thank you, we appreciate your feedback and we agree this should be addressed in the discussion. The following text elaborates on the cohort size for the studies, “The cohorts used in these studies had limited patient numbers and it is possible that a future larger scale evaluation would further refine, update, or alter the findings of these pilot studies.”

Reviewer 2, point 21: “The quality of the graphs is bad, but I am not sure if this is due to the journal, provided that this is review material. If this is the resolution of the original images, then they should be improved prior to potential publication.”

Response: Thank you. We have regenerated high resolution images for the figures.

We would like to thank the reviewers for their thoughtful comments. We believe that we have addressed every single comment the reviewers have brought up.

---

## [Decision Letter · Decision Letter 1]

29 Aug 2023

Integrating current analyses of the breast cancer microbiome

PONE-D-23-13336R1

Dear Dr. Sohail,

We’re pleased to inform you that your manuscript has been judged scientifically suitable for publication and will be formally accepted for publication once it meets all outstanding technical requirements.

Kind regards,

Rajesh P. Shastry, Ph.D

Academic Editor

PLOS ONE

Additional Editor Comments (optional):

Reviewers' comments:

Reviewer's Responses to Questions

**Comments to the Author**

1. If the authors have adequately addressed your comments raised in a previous round of review and you feel that this manuscript is now acceptable for publication, you may indicate that here to bypass the “Comments to the Author” section, enter your conflict of interest statement in the “Confidential to Editor” section, and submit your "Accept" recommendation.

Reviewer #2: All comments have been addressed

2. Is the manuscript technically sound, and do the data support the conclusions?

Reviewer #2: Yes

3. Has the statistical analysis been performed appropriately and rigorously? 

Reviewer #2: N/A

4. Have the authors made all data underlying the findings in their manuscript fully available?

Reviewer #2: Yes

5. Is the manuscript presented in an intelligible fashion and written in standard English?

Reviewer #2: Yes

6. Review Comments to the Author

Reviewer #2: (No Response)

7. PLOS authors have the option to publish the peer review history of their article (what does this mean?). If published, this will include your full peer review and any attached files.

Reviewer #2: No

---

## [Editor Report · Acceptance letter]

4 Sep 2023

PONE-D-23-13336R1 

Integrating current analyses of the breast cancer microbiome 

Dear Dr. Sohail:

I'm pleased to inform you that your manuscript has been deemed suitable for publication in PLOS ONE. Congratulations! Your manuscript is now with our production department. 

Kind regards, 

on behalf of

Dr. Rajesh P. Shastry 

Academic Editor

PLOS ONE